# The Impact of Telemedicine on Parkinson’s Care during the COVID-19 Pandemic: An Italian Online Survey

**DOI:** 10.3390/healthcare10061065

**Published:** 2022-06-08

**Authors:** Fabiana Ruggiero, Linda Lombi, Maria Takeko Molisso, Giorgio Fiore, Eleonora Zirone, Roberta Ferrucci, Elena Pirola, Marco Locatelli, Sergio Barbieri, Francesca Mameli

**Affiliations:** 1Foundation IRCCS Ca’ Granda Ospedale Maggiore Policlinico, 20122 Milan, Italy; maria.molisso@policlinico.mi.it (M.T.M.); giorgio.fiore@unimi.it (G.F.); eleonora.zirone@policlinico.mi.it (E.Z.); elena.pirola@policlinico.mi.it (E.P.); marco.locatelli@policlinico.mi.it (M.L.); sergio.barbieri@policlinico.mi.it (S.B.); francesca.mameli@policlinico.mi.it (F.M.); 2Department of Sociology, Università Cattolica del Sacro Cuore, 20125 Milan, Italy; linda.lombi@unicatt.it; 3Department of Health Sciences, “Aldo Ravelli” Center for Neurotechnology and Experimental Brain Therapeutics, University of Milan, 20142 Milan, Italy; roberta.ferrucci@unimi.it

**Keywords:** survey, Parkinson’s disease, COVID-19, telehealth, telemedicine, E-health, telerehabilitation

## Abstract

Traditionally, medical care and research in Parkinson’s disease (PD) have been conducted through in-person visit. The recent Coronavirus Disease 2019 (COVID-19) pandemic has profoundly impacted the delivery of in-person clinical care. We conducted an online survey to investigate the impact of COVID-19 on access to telehealth care, interviewing both PD patients and neurologists. Survey responses were collected from 1 March to 31 May 2021 through an anonymous, self-reported questionnaire, on the ‘Qualtrics’ platform. In total, 197 patients and 42 neurologists completed the survey. In our sample, 37.56% of PD patients and 88.10% of neurologists reported having used alternatives to in-person visits, while 13.70% of PD patients and 40.48% of neurologists used telemedicine. Data showed that respondents were generally satisfied with the use of telemedicine during the COVID-19 pandemic. The relational dimension between patient and neurologist seems to be the factor that most positively affected the telemedicine experience, contributing greatly to a more patient-centred care. Current findings suggest the need to improve the access to telehealth services for patients with PD. The technology has the potential to improve the care of frail patients, especially when availability of face-to-face visits is limited.

## 1. Introduction

The Coronavirus Disease 2019 (COVID-19) pandemic had a severe impact on our society, disrupting the lives and routines of many people worldwide and deeply affecting their lifestyle and physical and mental well-being. To contain the spread of the COVID-19 infection, nations around the world adopted restrictive measures, including social distancing, home confinement, and lockdown. In Europe, Italy was the first country to declare a national lockdown, on 11 March 2020 [1].

The impact might have been even stronger in people already vulnerable due to chronic conditions [2], such as Parkinson’s disease (PD) and other movement disorders [3,4]. PD is a neurodegenerative disease involving motor disorders but also psychological and cognitive symptoms, the most common of which are depression, apathy, and cognitive impairment [5].

The pandemic and the containment measures may have also had an indirect and negative impact on healthcare provided to PD patients. The surge of COVID-19 cases proved to be extremely challenging for healthcare facilities, forced to re-organize hospital activities [6,7]. During the lockdown, outpatients’ in-person medical visits were disrupted, as was the delivery of rehabilitation programs [8]. Furthermore, a previous study indicated that several PD patients spontaneously gave up visits or rehabilitation for fear of contracting the virus in hospitals or rehabilitation facilities [9]. On this background, to provide continuous care for movement disorders, in-person visits were occasionally and temporarily replaced by alternatives, such as telehealth.

Telehealth, which is the use of electronic information and telecommunication technologies to support and promote long-distance clinical health care [10], played an increasingly central role over the course of the pandemic [11]. Telehealth services allow health professionals to manage their patients remotely, by administering virtual follow-up visits and monitoring the onset of new symptoms or emergencies, while patients receive continuous care support; therefore, these services represented a ‘first line’ tool during the pandemic. Recently, telehealth aimed not only at caring for patients through remote consultation, replacing face-to-face intervention. Actually, it might be useful to reduce hospitalization rates and provide specific immediate home-based services, guaranteeing continuous monitoring of patients and improving not only the state of health but also above all the quality of life [12]. However, there are obstacles preventing a wide use of telehealth, such as patients’ and healthcare professionals’ poor technology skills, insufficient patient compliance during virtual visits [13], or unavailability of digital devices. Although many studies on the use of telehealth have been recently produced, partly due to the impetus provided by the pandemic, the literature on the use of telehealth in PD is still limited [14]. It is, therefore, essential to assess how telehealth is currently used by patients and healthcare professionals.

This study aims to evaluate the critical issues encountered by both PD patients and neurologists, in order to outline new socio-health care models to meet the needs of care even in complex conditions such as a pandemic.

## 2. Materials and Methods

We administered an online survey to PD patients and movement disorders specialists, to investigate their experiences of remote neurological consultations with a specific focus on its use during the pandemic.

Survey responses were collected from 1 March to 31 May 2021. To reach a wider audience we adopted a convenience sample.

The survey was promoted on social media and sent via e-mail to both associations of patients with movement disorders and neurology outpatient clinics in Italy. Data collection was conducted through an anonymous, structured, self-reported questionnaire, administered online through the ‘Qualtrics’ platform (www.qualtrics.com, accessed on 1 March 2021). ‘Qualtrics’ is an online survey platform used to create customized surveys with a drag-and-drop interface.

The appropriate Ethics Committee reviewed and approved survey dissemination for this study in accordance with local legal standards and authorized disclosure (Protocol no. 184_2021bis). Because of the pandemic, the consent to personal data processing was provided online by participants before responding to the survey.

### 2.1. Survey Development and Structure

The survey was created by researchers with clinical psychological expertise in the field of movement disorders and with competence in the sociology of health. All questions included in the survey were developed on existing literature on telehealth research [15], in order to identify the literature gaps on the less-investigated aspects, mainly in terms of impact (see Appendix A).

The survey consisted of 3 sections: a first section collecting demographic and clinical data; a second section collecting data on the use of different technological and digital alternatives to in-person visits, before and during the pandemic; and a third section accessible only to users of video consultations. For the purpose of this study, video consultation is defined as ‘telemedicine’, understood as the practice of medicine using electronic communication, information technology, or other means between a physician in one location and a patient in another location [16].

This last section included a series of statements aimed at examining the following constructs: ‘Satisfaction’, ’Experience’, ‘Technical Quality’, ‘Effectiveness’, ‘Usefulness/Perceived Usefulness’ and ‘Effect on Interaction’ [15] (Table 1).

Participants were asked whether they agreed with several statements, rating their agreement on a Likert scale ranging from 0 (‘strongly disagree’) to 4 (‘strongly agree’).

### 2.2. Statistical Analysis

Editing and sorting processes were performed using Microsoft Excel. Descriptive statistics were applied to the data. Mean and standard deviations were used for continuous variables, while frequency and percentage were calculated for categorical variables. Statistical analysis of data was performed using TIBCO Statistica™ software (version 13.5, Palo Alto, CA, USA). Spearman’s rank correlation coefficient (ρ) was calculated to assess the presence of correlation between scores of survey constructs.

Correlation between each construct and all other constructs was calculated. We focused on high (ρ = 0.7–0.9) and very high (ρ = 0.9–1) positive and negative correlations between the constructs [17]. Statistical significance was set at *p* < 0.05.

## 3. Results

### 3.1. PD Patients

A total of 197 PD patients fully completed the survey (out of 276 who started the survey, the completion rate 71.38%).

Among the respondents, 170 reported that they never used alternatives to in-person visits before the pandemic (86.29%). One hundred twenty-three (62.44%) reported that they did not use any alternative to in-person visit during the pandemic, while 74 (37.56%) had. Twenty-seven of them (13.71%) used video consultation, but only 20 patients (10.15%) provided information on the constructs investigated in this study (Table 2).

#### 3.1.1. Alternatives to In-Person Visits

Seventy-four PD patients (37.56%) (33 men; mean ± SD; age: 63.11 ± 10.15 years; education: 13.72 ± 3.96 years; disease duration: 9.65 ± 6.24 years) reported having used alternatives to in-person visits: 34 of them used e-mail (45.94%), 27 video consultation (36.49%), 10 telephone calls (13.52%), while use of SMS and chat services were less frequent in our sample (SMS: 2 patients (2.70%); chat: 1 patient (1.35%).

Forty-five (60.81%) of the participants were retired, while 17 were employed (22.97%), 10 were unemployed (13.52%), and 2 did not provide information on their working status (2.70%).

Forty-six patients reported having been informed of the availability of remote neurological consultation by their physician (62.16%), 12 patients were made aware by other information channels (16.22%), 9 by patient associations (12.16%), 5 by Internet sources (6.76%), 1 by other patients (1.35%), while 1 patient did not report the source.

A neurological consultation was necessary because of the following reasons: follow-up visit (23 patients; 31.08%), drugs prescription (11 patients, 14.86%), drugs side-effects (6 patients; 8.12%), other enquiries (8 patients; 10.81%), DBS parameters regulation (2 patients; 2.70%), concurrence of more than one of the aforementioned needs (22 patients; 29.73%). Two patients (2.70%) did not disclose the reason.

Forty-seven respondents did not require technological assistance for the setting up of the remote consultation (63.51%) (Table 2).

#### 3.1.2. Telemedicine

Twenty-seven PD patients (13.71%) reported using video consultation, but only 20 of them (10.15%) filled in the questions on the indicators of telemedicine use (10 men; mean ± SD; age: 60.75 ± 11.12 years; education: 14.5 ± 3.28 years; disease duration: 9.05 ± 5.58 years).

In our sample, 45% of the respondents to the constructs reported having slight symptoms (minimal impact of the disease or only one side of the body affected), while 90% reported having no comorbidities; 50% of the patients reported being retired, while 30% were employed and 20% unemployed; 55% of patients reported living in Northern Italy, while 45% in Central-Southern Italy (Table 2).

According to the responses collected on the constructs investigated by the survey, patients reported for ‘Perceived Utility’ that the video consultation was overall useful (65% very useful, 25% quite useful, 5% neither useful nor useless, while 5% found it not very useful). Regarding ‘Effectiveness’, video consultation was mostly perceived as effective by our sample (45% very effective, 40% quite effective, while 15% did not express a preference). When evaluating ‘Effect on interaction’, 55% of patients reported excellent quality of the interaction, 35% good quality, 5% did not express a preference, while the interaction was perceived as poor by 5%. Moreover, for the ‘Experience’ construct, 55% of patients ranked the experience as very good, 35% as good, while 10% did not express a preference. With regard to ‘Satisfaction’, 50% of patients reported being completely satisfied, 40% quite satisfied, while 10% did not express a preference. Finally, in regard to ‘Technical Quality’, 55% of patients reported excellent quality, 35% good quality, and 10% did not express a preference (Table 3).

Finally, correlations were highlighted by Spearman coefficient calculation between ‘Satisfaction’ and ‘Experience’ (ρ = 0.89, *p* < 0.001) and for ‘Interaction Effect’ (ρ = 0.89, *p* < 0.001).

The ‘Effectiveness’ construct was correlated with ‘Interaction Effect’ (ρ = 0.84, *p* < 0.001) and ‘Experience’ (ρ = 0.84, *p* < 0.001) and moderately correlated with ‘Satisfaction’ (ρ = 0.74, *p* < 0.001). Finally, a correlation was found between ‘Interaction Effect’ and ‘Experience’ (ρ = 0.99; *p* < 0.001) (Table 4).

### 3.2. Neurologists

A total of 42 neurologists fully completed the survey (out of the 62 who started the survey; completion rate 67.74%).

Among the respondents, 12 reported that they never used alternatives to in-person visits before the pandemic (28.57%). Thirty-seven neurologists (88.10%) reported using alternatives to in-person visit during the pandemic, of which 17 (40.48%) used video consultation and provided information on the constructs investigated in this study (Table 5).

#### 3.2.1. Alternatives to In-Person Visits

Thirty-seven neurologists (88.10%) (16 men; mean ± SD; age: 51.08 ± 11.15 years) reported having used alternatives to in-person visits: 17 respondents used video consultation (45.94%), 12 used e-mail (32.43%), 2 used telephone calls (5.41%), 1 used a chat service (2.70%), and 5 (13.51%) did not specify how they remotely interacted with their patients.

The option of resorting to a remote neurological consultation was suggested in most cases by the neurologists themselves (43.24%), but also by the hospital (37.84%) and by patients (13.51%), while 2 neurologists did not give such information.

Neurologists provided consultations only for follow-up visits (10.81%), or drugs prescriptions (5.41%), or drugs side effects (5.41%). No consultation was required only for DBS parameters regulation or only for other reasons. In most cases, however, a consultation was requested for a combination of the aforementioned reasons (64.86%).

In total 18.92% of respondents reported having received technical support for the setting up of remote neurological consultations (Table 5).

#### 3.2.2. Telemedicine

Seventeen neurologists (40.48%) reported using video consultation and provided information on the investigated constructs (8 men; mean ± SD; age: 52.47 ± 10.47 years).

Most of the respondents reported working in Northern Italy (82.35%) and having worked in public hospitals (58.82%) during the COVID-19 pandemic. About half of the participants (52.94%) used an institutional platform to deliver their video consultations (Table 5).

According to the responses collected on the constructs investigated by the survey, neurologists reported for ‘Perceived Utility’ that the video consultation was overall useful (23.53% very useful, 52.94% quite useful, 17.65% neither useful nor useless, 5.88% not very useful). Regarding ‘Effectiveness’, video consultation was mostly perceived as effective by our sample (23.53% very effective, 64.71% quite effective, while 1 neurologist (5.88%) did not provide an answer and 1 other (5.88%) did not find the video consultation very effective). When evaluating ‘Effect on interaction’, 11.76% of neurologists rated the interaction as excellent and 76.47% as quite good, while 5.88% did not answer and 5.88% of the participants rated the interaction as poor. Moreover, for the ‘Experience’ construct, 23.53% of neurologists reported that it was a very good experience, 52.94% as quite good, while 23.53% did not express a preference. With regard to the ‘Satisfaction’, 17.65% of neurologists reported that they were totally satisfied and 70.59% were quite satisfied, while 11.76% did not express a preference. Finally, regarding the ‘Technical Quality’, 23.53% of participants reported excellent technical quality of video consultations, 41.18% a good technical quality, while 23.53% did not express a preference and 11.76% reported poor technical quality (Table 3).

In addition, correlations between constructs were calculated. ‘Satisfaction’ construct was correlated with ‘Experience’ construct (ρ = 0.80; *p* < 0.001) and ‘Interaction Effect’ (ρ = 0.69, *p* = 0.002). ‘Usefulness/Perceived Usefulness’ construct was correlated with ‘Experience’ (ρ = 0.76, *p* < 0.001) and ‘Effectiveness’ (ρ = 0.75, *p* < 0.001). Finally, a positive correlation was found between ‘Interaction Effect’ and ‘Experience’ (ρ = 0.71, *p* = 0.002) (Table 4).

## 4. Discussion

In this study, we reported the results of an online survey aimed to assess the impact of COVID-19 on access to telehealth care for both PD patients and neurologists. In our sample, consisting of 197 PD patients and 42 neurologists, 62.43% of the PD patients answered that they were unable to access alternatives to the in-person neurological visit. The neurologists’ data were rather different: 88.09% of them resorted to telehealth to perform neurological consultations during the pandemic.

Moreover, clinicians reported having used alternatives to in-person neurological visits even before the COVID-19 pandemic, while 86% PD patients used them for the first time after the onset of the pandemic. This difference may be probably explained by the fact that, even before the pandemic, neurologists may have offered some form of virtual consultations to vulnerable patients already undergoing treatment or for clinical monitoring.

Nonetheless, traditionally medical care in PD have been conducted with in-person encounters probably to accurately assess their motor symptoms. In response to the health emergency, telehealth became the safest way to ensure continuity of care to patients, especially to those affected by chronic disabling diseases—such as PD—requiring frequent medical consultations and therapeutic adjustments [18]. The rapid spread of the virus and concomitant lack of resources slowed down the digitization process of the Italian health care system, which was not already equipped to support a transition to virtual care [19]. Other factors that may have prevented a rapid transition to telehealth services were the inadequacy of regulations in place for the delivery of care through digital systems, and the socio-cultural background: the pandemic-related crisis has certainly given new impetus to the transition process [20,21].

In our sample, 37.56% of PD patients and 88.10% of neurologists reported having used alternatives to in-person neurological visits, while 13.70% PD patients and 40.48% of neurologists used video consultation. The most popular tools used by both neurologists and patients to communicate remotely were e-mail (32.43% of neurologists, 45.94% of patients) and video consultation (40.48% of neurologists, 36.49% of patients): quite often (31.08%) the reason behind a medical consultation was to receive advice that clinicians were able to provide by e-mail, while more complex clinical monitoring was performed through video consultation, a methodology that allows physicians to check on the patients’ clinical and motor conditions.

Although some steps of the neurologic examination cannot be performed remotely (e.g., pull test and muscle tone assessment), most of the movement disorders assessment is suitable for an audio-visual approach, using the modified version of the Unified Parkinson’s Disease Rating Scale (UPDRS) as a validated proxy for the full motor assessment [22]. Moreover, in order to support movement disorders neurologists worldwide in dealing with telemedicine, the Movement Disorders Society Telemedicine Study Group published a step-by-step guide [23], which includes references to specific regulations and requirements for reimbursement based on the latest information available at regional and national levels.

In our study, we also investigated how the use of telemedicine affected the doctor-patient relationship as well as the impact of telemedicine on the health care systems, in terms of efficiency, costs, satisfaction, and involvement of patients and doctors. To do so, we developed a questionnaire with specific constructs identified through a literature review [15]. There was a consensus with participants that telemedicine should play a role in the delivery of care to patients with PD while the pandemic lasts and that patient and neurologist satisfaction may be improved by a higher technical quality of the examination.

Benefits are largely accrued by patients, and they are both tangible (reduced travel time and costs to access the specialist) and intangible (patient convenience, video consultation to prevent disruption of care, and good communication). The relational dimension, as expressed by patients’ and doctors’ reported experience, is the factor that most positively correlates with the level of Experience. In fact, the doctor–patient relationship may be one of the most complex interpersonal relationships, being emotionally loaded and requiring close mutual cooperation to reach a shared goal. Although the classical setting of examinations has radically changed with the advent of video consultation, telemedicine providers felt that video communication does not have a negative impact on doctor–patient relationships. Moreover, our data suggest that the distance between doctor and patient during a video consultation can be reduced by the doctor’s attitude and patient’s approach. In fact, patients reported that the neurologist’s attitude during the consultation is crucial to convey willingness to listen and empathy.

The Experience construct sheds light on patients’ experience while receiving care and can play a key role in offering a more patient-centred health care delivery and in improving the safety and quality of care. Results from our survey show an overall appreciation of a more patient-centred health care system, which may also facilitate the building of a doctor–patient partnership. These results suggest that a new approach to the delivery of care may support the patient, now actively included in the medical process, and the clinicians as well, which may get a better understanding of their patients’ preferences, values, and psycho-physiological comfort; thus, flexibility in administering health care may move beyond the traditional paternalistic approach.

Moreover, our results highlight the quality of health care services that are tailored to the patient’s needs through telemedicine improved. In a virtuous cycle, the shift to a patients-centred care approach also improved concordance between health care providers, patients’ adherence to treatment plans [24,25], and health outcomes, while at the same time it increased patients’ satisfaction with healthcare services [26,27].

Our data highlighted the importance of the communication process, in which evaluation of possible complications and treatment outcomes improve when the diagnosis is more clearly communicated, and the patient is more actively involved in the management and treatment of their illness. In line with previous observations, our results show that doctor–patient communication can significantly influence the attitude of patients towards their well-being, their understanding of medical information, the perception of their disease, their quality of life, and their health status.

### Limitations

There are several limitations in this study, thus our findings should be interpreted with caution.

The first limitation is the low response rate: a low response rate in a descriptive study presents a generalization of the findings, which cannot be applied to a broader segment of PD patients. Moreover, adopting a convenience sample, the survey was conducted online, and shared on social media and by e-mail of patient associations: therefore, the answers may not be fully representative of the Italian population of PD patients, as they do not necessarily reflect the view of patients with limited or no access to information technology. Furthermore, the survey mainly reached patients who are part of networks or members of patient organizations, who are likely to be more involved in their own treatment and care; once again, their answers may not reflect the views of the wider population of patients.

Due to the wide distribution of the survey on the Internet, we are also unable to ascertain the response rate compared to the number of patients who had access to the survey itself, and whether there were systematic differences between respondents who completed the survey and those who did not.

Furthermore, disease duration, which was not assessed based on clinical data, may have impacted the satisfaction of virtual consultation: recently diagnosed patients may not have yet developed a relation of trust with their physicians, and the resulting lack of confidence may have negatively affected the rating of virtual and video consultations in terms of satisfaction.

For the purpose of this study, we collected data from both experienced and inexperienced telehealth service users. Most PD patients have scarce experience with telemedicine services designed for movement disorder management; therefore, reaching experienced users was challenging; future studies should focus on PD patients that already have experience with telemedicine, which will soon become more and more widespread.

Regarding the neurologists’ group, it is worth noting that the respondents are movement disorders specialists, whose opinions may reflect personal biases developed during previous clinical experiences: consequently, the survey’s findings may not apply to other neurological or medical specialists.

A wider survey taking these factors into account may result in different data and findings. More specifically, a similar survey repeated at the end of the pandemic would allow to determine whether and how current views on virtual consultations were affected by the necessity of resorting to alternatives to in-person visits while COVID-19 restrictions were in place.

## 5. Conclusions

Our findings provide baseline information to design improved telemedicine services, which can open paths to reach the goal of a user-friendly and satisfactory experience of virtual care delivery for movement disorders. The benefits of telemedicine are several and promising, and it has several possible applications in PD management: we believed that the application of technology in medicine is useful for patients, for healthcare providers, and institutions, allowing easy access to medical service, avoiding stressful travels, and reducing the health care costs and waiting lists. Future studies should be addressed in this landscape to improve the quality of service, discover the best-applicated use, and spread it among patients.

## Figures and Tables

**Table 1 healthcare-10-01065-t001:** List of indicators, definitions, and variables.

Indicators	Definition	Variables
satisfaction	subjective evaluation of whether the user’s expectations were met	-overall satisfaction -willingness to use/re-use
experience	evaluation of the user’s experience of a healthcare service: the experience can be objective (e.g., waiting time) or subjective (e.g., level of patient-centredness)	-patient-reported experience measures -comfort -patient-centredness
technical quality	subjective evaluation of the quality of the technology used	-audio and picture (video) -quality -reliability -usability/ease of use -intuitiveness/learnability -privacy and security
effectiveness	objective/subjective assessment that a telehealth interaction helped improve the health status or well-being of a patient	-quality of life -change in health status -measures of health/well-being -patient empowerment -patient knowledge -patient-reported outcome measures
usefulness/perceived usefulness	objective or subjective assessment that a telehealth interaction produced some benefit or met the purpose of the interaction	-convenience -time consequences -cost consequences -accessibility -effect on continuity of care -(future) intention to use -willingness to use/practice -acceptability
effect on interaction	subjective assessment that the modality of communication affected clinician–patient or clinician–clinician interaction	-communication style -ease of communication -completeness of information

**Table 2 healthcare-10-01065-t002:** Demographic data of PD patients and telehealth groups.

Demographic Data		197 Survey Respondents	74 Alternatives to In-Person Visits Group	27 Video Consultation Group	20 Respondents to Video Consultation Constructs
		Mean ± SD	Mean ± SD	Mean ± SD	Mean ± SD
age (years)		65.65 ± 10.26	63.11 ± 10.15	60.63 ± 10.78	60.75 ± 11.12
education (years)		13.49 ± 3.93	13.72 ± 3.96	14.11 ± 3.75	14.5 ± 3.28
disease duration (years)		9.67 ± 6.66	9.65 ± 6.24	9.96 ± 7.36	9.05 ± 5.58
		*n* (%)	*n* (%)	*n* (%)	*n* (%)
gender	male	97 (49.24)	33 (44.60)	12 (44.44)	10 (50)
	female	100 (50.76)	41 (55.40)	15 (55.56)	10 (50)
occupation	unemployed	12 (6.09)	10 (13.52)	5 (18.52)	4 (20)
	working	46 (23.35)	17 (22.97)	7 (25.93)	6 (30)
	retired	134 (68.02)	45 (60.81)	14 (51.85)	10 (50)
	other	5 (2.54)	2 (2.70)	1 (3.70)	0 (0)
area of residence	northern Italy	120 (60.91)	44 (59.46)	16 (59.26)	11 (55)
	central-southern Italy	77 (39.09)	30 (40.54)	11 (40.74)	9 (45)
home assistance	no	21 (10.66)	11 (14.86)	5 (18.52)	5 (25)
	yes	176 (89.34)	63 (85.14)	22 (81.48)	15 (75)
disease diagnosis	Parkinson’s disease	168 (85.28)	67 (90.54)	27 (96.30)	19 (95)
	atypical parkinsonisms	29 (14.72)	7 (9.46)	0 (0)	1 (5)
disease grade	slight	48 (24.36)	22 (29.73)	11 (40.74)	9 (45)
	mild	27 (13.71)	14 (18.92)	6 (22.22)	4 (20)
	moderate	60 (30.46)	18 (24.32)	6 (22.22)	4 (20)
	moderate-severe	44 (22.33)	15 (20.27)	2 (7.41)	1 (5)
	severe	18 (9.14)	5 (6.76)	2 (7.41)	2 (10)
comorbidity	psychiatric symptoms	13 (6.60)	3 (4.05)	0 (0)	0 (0)
	cognitive impairment	14 (7.11)	5 (6.76)	2 (7.41)	1 (5)
	none	162 (82.23)	64 (86.49)	23 (85.19)	18 (90)
	psychiatric symptoms and cognitive impairment	2 (1.02)	1 (1.35)	1 (3.70)	1 (5)
	na	6 (3.04)	1 (1.35)	1 (3.70)	0 (0)
previous use of telehealth	no	170 (86.29)	51 (68.92)	19 (70.37)	15 (75)
	yes	27 (13.71)	23 (31.08)	8 (29.63)	5 (25)
use of telehealth during COVID-19	no	123 (62.44)	0 (0)	0 (0)	0 (0)
	yes	74 (37.56)	74 (100)	27 (100)	20 (100)
telehealth technical support	no	47 (23.86)	47 (63.51)	14 (51.85)	13 (65)
	yes	26 (13.20)	26 (35.14)	13 (48.15)	7 (35)
	na	124 (62.94)	1 (1.35)	0 (0)	0 (0)
main reason(s) for consultation	follow up	23 (11.68)	23 (31.08)	11 (40.74)	7 (35)
	drug prescriptions	11 (5.58)	11 (14.86)	5 (18.52)	4 (20)
	DBS parameters regulation	2 (1.02)	2 (2.70)	0 (0)	0 (0)
	side-effect to drugs	6 (3.04)	6 (8.12)	4 (14.81)	3 (15)
	other	8 (4.06)	8 (10.81)	3 (11.12)	3 (15)
	mixed condition	22 (11.17)	22 (29.73)	4 (14.81)	3 (15)
	na	125 (63.45)	2 (2.70)	0 (0)	0 (0)
telehealth suggested by	doctor	46 (23.35)	46 (62.16)	20 (74.07)	15 (75)
	patient’s associations	9 (4.57)	9 (12.16)	5 (18.52)	3 (15)
	patients	1 (0.51)	1 (1.35)	0 (0)	0 (0)
	internet	5 (2.54)	5 (6.76)	0 (0)	0 (0)
	other	12 (6.09)	12 (16.22)	2 (7.41)	2 (10)
	na	124 (62.94)	1 (1.35)	0 (0)	0 (0)
telehealth modality	none	123 (62.44)	0 (0)	0 (0)	0 (0)
	telephone	10 (5.07)	10 (13.52)	0 (0)	0 (0)
	sms	2 (1.02)	2 (2.70)	0 (0)	0 (0)
	chat	1 (0.51)	1 (1.35)	0 (0)	0 (0)
	e-mail	34 (17.25)	34 (45.94)	0 (0)	0 (0)
	video consultation	27 (13.71)	27 (36.49)	27 (100)	20 (100)

Legend. na: data not available.

**Table 3 healthcare-10-01065-t003:** Construct response data of PD patients and neurologists.

		20 Patients	17 Neurologists
		*n* (%)	*n* (%)
perceived usefulness	strongly disagree	0 (0)	0 (0)
	quite in disagreement	1 (5)	1 (5.88)
	neither in agreement nor disagreement	1 (5)	3 (17.65)
	quite in agreement	5 (25)	9 (52.94)
	strongly agree	13 (65)	4 (23.53)
technical quality	strongly disagree	0 (0)	1 (5.88)
	quite in disagreement	0 (0)	1 (5.88)
	neither in agreement nor disagreement	2 (10)	4 (23.53)
	quite in agreement	7 (35)	7 (41.18)
	strongly agree	11 (55)	4 (23.53)
effectiveness	strongly disagree	0 (0)	0 (0)
	quite in disagreement	0 (0)	1 (5.88)
	neither in agreement nor disagreement	3 (15)	1 (5.88)
	quite in agreement	8 (40)	11 (64.71)
	strongly agree	9 (45)	4 (23.53)
effect on interaction	strongly disagree	0 (0)	0 (0)
	quite in disagreement	1 (5)	1 (5.88)
	neither in agreement nor disagreement	1 (5)	1 (5.88)
	quite in agreement	7 (35)	13 (76.47)
	strongly agree	11 (55)	2 (11.76)
experience	strongly disagree	0 (0)	0 (0)
	quite in disagreement	0 (0)	0 (0)
	neither in agreement nor disagreement	2 (10)	4 (23.53)
	quite in agreement	7 (35)	9 (52.94)
	strongly agree	11 (55)	4 (23.53)
satisfaction	strongly disagree	0 (0)	0 (0)
	quite in disagreement	0 (0)	0 (0)
	neither in agreement nor disagreement	2 (10)	2 (11.76)
	quite in agreement	8 (40)	12 (70.59)
	strongly agree	10 (50)	3 (17.65)

**Table 4 healthcare-10-01065-t004:** Correlations between constructs for PD patients and neurologists.

	Usefulness/ Perceived Usefulness	Technical Quality	Effectiveness	Effect on Interaction	Experience	Satisfaction
	20 patients					
usefulness/perceived usefulness	-	ρ = 0.67 *p* = 0.001	ρ = 0.67 *p* = 0.001	ρ = 0.67 *p* = 0.001	ρ = 0.67 *p* = 0.001	ρ = 0.61 *p* = 0.004
technical quality	ρ = 0.67 *p* = 0.001	-	ρ = 0.49 *p* = 0.029	ρ = 0.54 *p* = 0.014	ρ = 0.54 *p* = 0.014	ρ = 0.58 *p* = 0.007
effectiveness	ρ = 0.67 *p* = 0.001	ρ = 0.49 *p* = 0.029	-	ρ = 0.84 * *p* < 0.001	ρ = 0.84 * *p* < 0.001	ρ = 0.74 * *p* < 0.001
effect on interaction	ρ = 0.67 *p* = 0.001	ρ = 0.54 *p* = 0.014	ρ = 0.84 * *p* < 0.001	-	ρ = 0.99 * *p* < 0.001	ρ = 0.89 * *p* < 0.001
experience	ρ = 0.67 *p* = 0.001	ρ = 0.54 *p* = 0.014	ρ = 0.84 * *p* < 0.001	ρ = 0.99 * *p* < 0.001	-	ρ = 0.89 * *p* < 0.001
satisfaction	ρ = 0.61 *p* = 0.004	ρ = 0.58 *p* = 0.007	ρ = 0.74 * *p* < 0.001	ρ = 0.89 * *p* < 0.001	ρ = 0.89 * *p* < 0.001	-
	17 neurologists					
usefulness/perceived usefulness	-	ρ = 0.76 * *p* < 0.001	ρ = 0.75 * *p* < 0.001	ρ = 0.50 *p* = 0.041	ρ = 0.76 * *p* < 0.001	ρ = 0.65 *p* = 0.004
technical quality	ρ = 0.76 * *p* < 0.001	-	ρ = 0.62 *p* = 0.008	ρ = 0.37 *p* = 0.143	ρ = 0.57 *p* = 0.017	ρ = 0.41 *p* = 0.105
effectiveness	ρ = 0.75 * *p* < 0.001	ρ = 0.62 *p* = 0.008	-	ρ = 0.41 *p* = 0.103	ρ = 0.73 * *p* = 0.001	ρ = 0.37 *p* = 0.138
effect on interaction	ρ = 0.50 *p* = 0.041	ρ = 0.37 *p* = 0.143	ρ = 0.41 *p* = 0.103	-	ρ = 0.71 * *p* = 0.002	ρ = 0.69 * *p* = 0.002
experience	ρ = 0.76 * *p* < 0.001	ρ = 0.57 *p* = 0.017	ρ = 0.73 * *p* = 0.001	ρ = 0.71 * *p* = 0.002	-	ρ = 0.80 * *p* < 0.001
satisfaction	ρ = 0.65 *p* = 0.004	ρ = 0.41 *p* = 0.105	ρ = 0.37 *p* = 0.138	ρ = 0.69 * *p* = 0.002	ρ = 0.80 * *p* < 0.001	-

Legend. Spearman’s rank coefficient ρ ≥ 0.7 and *p* < 0.05. *: indicate statistical significance.

**Table 5 healthcare-10-01065-t005:** Demographic data of neurologist and telehealth groups.

Demographic Data		42 Survey Respondents	37 Alternatives to In-Person Visits Group	17 Video Consultation Group
		Mean ± SD	Mean ± SD	Mean ± SD
age (years)		50.55 (12.52)	51.08 (11.15)	52.47 (10.47)
		*n* (%)	*n* (%)	*n* (%)
gender	male	19 (45.24)	16 (43.24)	8 (47.06)
	female	23 (54.76)	21 (56.76)	9 (52.94)
working region	northern Italy	31 (73.81)	28 (75.68)	14 (82.35)
	central-southern Italy	11 (26.19)	9 (24.32)	3 (17.65)
working place	public hospital	22 (52.38)	21 (56.76)	10 (58.82)
	private hospital	4 (9.52)	4 (10.81)	2 (11.76)
	private practice	3 (7.14)	3 (8.11)	0 (0)
	university	5 (11.91)	3 (8.11)	1 (5.88)
	other	8 (19.05)	6 (16.22)	4 (23.53)
working experience (years)	0–5	6 (14.28)	4 (10.81)	1 (5.88)
	6–15	11 (26.19)	11 (29.73)	5 (29.41)
	16–25	10 (23.81)	9 (24.32)	4 (23.53)
	26–35	8 (19.05)	7 (18.92)	5 (29.41)
	>35	7 (16.67)	6 (16.22)	2 (11.76)
previous use of telehealth	no	12 (28.57)	7 (18.92)	2 (11.76)
	yes	30 (71.43)	30 (81.08)	15 (88.24)
use of telehealth during COVID-19	no	5 (11.90)	0 (0)	0 (0)
	yes	37 (88.10)	37 (100)	17 (100)
telehealth suggested by	hospital	14 (33.34)	14 (37.84)	7 (41.18)
	themselves	16 (38.09)	16 (43.24)	7 (41.18)
	patient	5 (11.90)	5 (13.51)	3 (17.65)
	na	7 (16.67)	2 (5.41)	0 (0)
institutional platform	no	23 (54.76)	23 (62.16)	9 (52.94)
	yes	12 (28.57)	12 (32.43)	8 (47.06)
	na	7 (16.67)	2 (5.41)	0 (0)
telehealth technical support	no	4 (9.52)	4 (10.81)	3 (17.65)
	yes	7 (16.67)	7 (18.92)	5 (29.41)
	na	31 (73.81)	26 (70.27)	9 (52.94)
institutional guidelines	no	6 (14.28)	6 (16.22)	5 (29.41)
	yes	5 (11.90)	5 (13.51)	3 (17.65)
	na	31 (73.81)	26 (70.27)	9 (52.94)
main reason(s) for consultation	follow up	4 (9.52)	4 (10.81)	3 (17.65)
	drug prescriptions	2 (4.76)	2 (5.41)	1 (5.88)
	DBS parameters regulation	0 (0)	0 (0)	0 (0)
	side-effect to drugs	2 (4.76)	2 (5.41)	0 (0)
	other	0 (0)	0 (0)	0 (0)
	mixed condition	24 (57.14)	24 (64.86)	13 (76.47)
	na	10 (23.82)	5 (13.51)	0 (0)
telehealth modality	none	0 (0)	0 (0)	0 (0)
	telephone	2 (4.76)	2 (5.41)	0 (0)
	sms	0 (0)	0 (0)	0 (0)
	chat	1 (2.38)	1 (2.70)	0 (0)
	e-mail	12 (28.57)	12 (32.43)	0 (0)
	video consultation	17 (40.48)	17 (45.94)	17 (100)
	na	10 (23.81)	5 (13.51)	0 (0)

Legend. na: data not available.

## Data Availability

The data presented in this study are available on request from the corresponding author.

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
