# Peer review of "The Impact of Telemedicine on Parkinson’s Care during the COVID-19 Pandemic: An Italian Online Survey"

_healthcare, 2022, doi:10.3390/healthcare10061065_

Round 1

Reviewer 1 Report

In this manuscript, Ruggiero F., et al. evaluate the critical issues encountered by both Parkinson's disease (PD) patients and neurologists, in order to outline new socio-health care models to meet the needs of care even in complex conditions such as the COVID-19 pandemic. They administered an online survey to PD patients and movement disorders specialists to investigate their experiences of remote neurological consultations with a specific focus on its use during the pandemic. A total of 197 PD patients and 42 neurologists fully completed the survey. Among the respondents, 74 (37.56%) PD patients and 37 (88.10%) neurologists used alternatives to in-person neurological visits during the pandemic. In their sample, 65% of PD patients reported that the video consultation was overall useful, while only 23.53% neurologists reported that the video consultation was overall useful. This manuscript highlights the importance of the communication process, in which technology has the potential to improve the care of frail patients, especially when availability of face-to-face visits is limited.

The scientific article is well written and described, the results are very interesting and explicative. All of the tables are appropriate in presenting the results, both in an easy and understandable way. However, minor changes are suggested:

1. In page 4 lines 116-117, where are the results of "only 20 patients (10.15%) provided information on the constructs investigated in this study" presented in Table 2?

2. In page 5 lines 141-144, where are the results of "only 20 of them (10.15%) filled in the questions on the indicators of telemedicine use" presented in Table 2?

3. In page 5 line 145, the manuscript mentions that 38.9% of the respondents reported having mild symptoms. However, in Table 2 the mild disease grade is shown as 27 (13.71%) or 6 (22.22%) for video consultation group. This need to be clarified.

4. In page 5 line 146-149, the manuscript mentions that 90% reported having no comorbidities; 50% of patients reported being retired, while 30% were employed and 20% unemployed; 55% of patients reported living in Northern Italy, while 45% in Central-Southern Italy. However, in Table 2 the values are different; 85.19% reported having no comorbidities; 51.85% of patients reported being retired, while 25.93% were employed and 18.52% unemployed; 59.26% of patients reported living in Northern Italy, while 40.74% in Central-Southern Italy. This need to be clarified.

5. In page 7 line 187, the manuscript mentions that 5.41% of patients suggested the option of resorting to a remote neurological consultation. However, in Table 3; 13.51% of patients suggested the option of resorting to a remote neurological consultation. This need to be clarified.

6. In page 7 line 189-191, the manuscript mentions that a consultation was requested for a combination of the aforementioned reasons and other needs, including adjustment of DBS parameters (64.86%). However, Table 3 shows that mixed condition is 64.86%, and DBS parameters was 0%. This need to be clarified.

7. The results written in the manuscript in Pages 5 lines 150-162 and Page 7 lines 201-216 can be presented in additional tables for better visualization and understanding.

Reviewer 2 Report

Comments – General

The study implemented an online survey approach to evaluate the self-reported use of telehealth technology by individuals with PD and neurologists as well as to examine their perceived experiences of using this modality of healthcare delivery. The authors did a commendable job of conducting a comprehensive assessment of the survey rating results. Some conceptual clarifications and minor concerns, however, need to be addressed.

Major Comments – Introduction

1.          Pg. 2 Lines 89–90: The definition of telemedicine is not confined to the involvement of a video modality for clinical consultation. Refer to the official definitions from government agencies and world organizations for the technical differences between telehealth and telemedicine and use either term as appropriate, rather than interchangeably, throughout the manuscript.

a.     https://www.cdc.gov/phlp/publications/topic/anthologies/anthologies-telehealth.html

b.     https://www.ncbi.nlm.nih.gov/books/NBK45440/

c.     Telemedicine is “the delivery of healthcare services, where distance is a critical factor, by all healthcare professionals using information and communications technologies for the exchange of valid information for diagnosis, treatment and prevention of disease and injuries, research and evaluation, and for the continuing education of healthcare providers, all in the interests of advancing the health of individuals and their communities”. (WHO Telematics, 1998)

Major Comments – Materials and Methods

2.          Pg. 3 Lines 104–106: What is the rationale for having “focused on” rho values only between 0.7 and 1.0, but mentioning that any rho value above 0.5 was considered, too?

Major Comments – Results

3.          Pg. 4 Lines 109–117: These summary results need not be described in text because readers can get the same information from the tables. Consider only including important information that is not available in direct form in the tables and/or briefly describe what each table contains.

4.          Pg. 6 Lines 164–169 and Pg. 8 Lines 218–221: For very small p values, indicate as “p < 0.001” to maintain consistent use of three significant figures for the reporting of results. Otherwise, if a p value can be rounded off to be a nonzero value, up to three decimal places, report this approximate value. For example, indicate as “p = 0.002” instead of “p = 0.0015”.

Major Comments – Discussion

5.          Pg. 9 Line 280: Elaborate on “distance”. Does it refer to physical distance or is it a figurative reference to the doctor-patient relationship?

6.          Pg. 9 Lines 293–294: What is the research evidence for this improvement. Cite reference(s) to substantiate this statement.

Minor Comments

7.          Pg. 3 Line 97: Do you mean “processes”?

8.          Pg. 3 Lines 97–99: Values of mean, standard deviation, frequency, and percentage are the results of analyzed data, not the data per se.

9.          Pg. 3 Line 103: This sentence does not contain sufficient content or refer to a standalone context to warrant a paragraph for itself; consider combining it with the following paragraph.

10.       Pg. 3 Line 104: It is redundant to specify “significant” because it can be assumed that only statistically significant results were reported.

11.       Use the written form instead of the spoken form for all words throughout the manuscript. For example, replace “didn’t” with “did not”.

12.       Pg. 4 Table 1 and Pg. 6 Table 2: Check consistency in using upper- versus lower-case for column headings.

Reviewer 3 Report

Dear Authors

The manuscript is not badly structured, it certainly lacks in methodological quality, so I recommend a major revision of the manuscript

L27 E-health, Telerehabilitation

L58 Perhaps we can broaden the discussion as follows:  “Recently, telehealth aimed not only at caring for patients through remote consultation, replacing face-to-face intervention. Actually, it might be useful to reduce hospitalization rates, provide specific immediate home-based services, guaranteeing continuous monitoring of patients and improving not only the state of health but also above all the quality of life.” (ref: https://doi.org/10.1108/JET-11-2020-0047 )

I suggest you follow the CROSS guidelines
https://www.researchgate.net/publication/351072130_A_Consensus-Based_Checklist_for_Reporting_of_Survey_Studies_CROSS

L65 The eligibility of the participants is totally missing, have you excluded people with cognitive deficits?

If the authors participated in the writing, add them in parentheses

Questionnaire, if possible, should be fully provided (in the article, or as appendices or as an online supplement).

ρ? For p value?

In results, enrich with information about the administration of the survey as per the guideline

Increase the quality of the table n = 71 for example, mean + -SD and n (%). Chi-square evaluations could be made

L221 I would represent these results with a table ..

The limitations section is very thick
